# First Detection of *Rhodococcus equi* in a Foal in Bulgaria—A Case Report

**DOI:** 10.3390/ani15142058

**Published:** 2025-07-12

**Authors:** Betina Boneva-Marutsova, Plamen Marutsov, Katharina Kerner, Georgi Zhelev

**Affiliations:** 1Department of Veterinary Microbiology, Infectious and Parasitic Diseases Faculty of Veterinary Medicine, Trakia University, 6000 Stara Zagora, Bulgaria; plamen.marutsov@trakia-uni.bg (P.M.); g.zhelev.georgiev@trakia-uni.bg (G.Z.); 2Laboklin GmbH & Co. KG, 97688 Bad Kissingen, Germany; kerner@laboklin.com

**Keywords:** *VapA*-positive *Rhodococcus equi*, foal, PCR, antimicrobial susceptibility

## Abstract

This report presents the first documented case of *VapA*-positive *Rhodococcus equi* infection in a foal in Bulgaria. The pathogen is known to often cause fatal pyogranulomatous bronchopneumonia and extrapulmonary disorders in foals, as well as in immunocompromised humans.

## 1. Introduction

*Rhodococcus equi* (formerly reclassified as *Rhodococcus hoagii*, = *R. hoagii*) is a ubiquitous aerobic, nonsporulating, nonmotile Gram-positive bacteria belonging to the genus Rhodococcus, Nocardiaceae family [1,2]. The primary virulence factor *of R. equi* is the virulence-associated protein A (*VapA*), which is encoded on a large plasmid. *VapA* plays a crucial role in enabling intracellular survival and evasion of host immune responses. Only *VapA*-positive strains have demonstrated the capacity to replicate within alveolar macrophages—an essential step in the development of pyogranulomatous bronchopneumonia in foals. Its expression is strongly correlated with pathogenicity. Consequently, PCR detection of the *VapA* gene not only confirms the identity of *R. equi* but also serves as a prognostic indicator of its virulence potential. The identification of *VapA*-positive strains is therefore critical for epidemiological surveillance, the implementation of targeted therapeutic protocols, and the development of effective prevention strategies [3,4]. *VapA*-positive *R. equi* strains are among the main causative agents of severe pyogranulomatous bronchopneumonia in foals [2,3]. The disease primarily affects young horses and is associated with high morbidity and mortality, causing serious health issues and economic losses to the equine industry [2]. Pyogranulomatous bronchopneumonia is the most common clinical manifestation; however, many affected foals may exhibit extrapulmonary disorders, including abdominal abscesses, colitis, osteomyelitis, and immune-mediated conditions such as nonseptic synovitis and uveitis [5]. The intracellular location of *R. equi*, its ability to survive and replicate in macrophages, the formation of abscesses with thick-walled capsules that impede the penetration of antimicrobial agents, and the tendency to develop resistance pose serious challenges in the treatment of affected individuals. This often results in a noticeable discrepancy between the in vitro activity of antimicrobial agents and their in vivo efficacy [2,6]. Control of the disease is also made more difficult by the infection of numerous domestic and wild animal species with an incompletely understood role in the epidemic process, but with potential roles in the transmission of the pathogen: dogs, cats, cattle, goats, sheep, lamas, camels, buffaloes, roe deer, deer, pigs, wild boars, and wild birds [2,7,8]. *R. equi* has gained attention as an opportunistic pathogen in human beings. Żychska et al. [2] note that the number of infections in immunocompromised people has been increasing in recent years, which is very worrying, especially due to the emerging antibiotic resistance.

The current article presents the case history of the first detection of *VapA*-positive *R. equi* in a foal in Bulgaria. The epidemiological and clinicopathological data, the diagnostic approach, and the antimicrobial susceptibility of the isolate are discussed.

## 2. Case Description

Stable information: A private stud farm stands a total of 55 horses, including 9 stallions, 30 mares (2 of which are still pregnant), 8 yearlings and older, and 8 foals.

Horse rearing: Summer—outdoor living 24/7 in paddocks. Winter—indoor housing in separated individual stalls. Middle seasons—outdoor in paddocks for 7–8 h per day if there is no rain, wind, or snow; indoor overnights.

Feeding: Horses are fed 10–12 kg good-quality hay, 2–4 kg alfalfa, and mineral supplements. Pregnant mares in the last trimester of gestation receive an additional 3–4 kg of concentrate.

Breeding management: The owner and manager of the horse farm have adopted a strategy of early foaling of the mares, and most of them give birth in the winter months. As a rule, the caretakers of newborns closely monitor when the foal stands up and begins nursing colostrum. This should ideally occur within two hours after birth. If the foal is unable to suckle on its own, colostrum is administered using an orogastric tube or through bottle feeding.

Vaccination and deworming schedule: Regular shots for equine influenza, rhinopneumonitis, and tetanus are applied annually. In the first year, foals receive three doses, starting at 6 months, then switch to a single booster dose annually. Regular deworming of all horses is carried out three times a year.

Case history: A 56-day-old female Arabian foal born on 22 January 2025, has been registered with the disease.

Physical examination: The previously healthy filly exhibited symptoms of depression, fever (40.8 °C), rapid breathing (46 breaths per minute), and an increased heart rate (82 beats per minute). Notably, there was significant respiratory distress, characterized by tachypnea, increased effort in breathing, increased coughing reflex, and flaring of the nostrils. Additionally, abdominal breathing and cyanosis of the mucous membranes were observed, along with a scant amount of muco-purulent nasal discharge from both nostrils. Auscultation of the lung field demonstrated crackles and bilateral wheezes. Milk intake was gradually decreased on day 3 and progressed to full inappetence. No other clinical symptoms were identified. The veterinarian’s presumed diagnosis was a common cold with subsequent pneumonia and bronchitis. Paraclinical tests such as blood tests, ultrasound, or X-rays were not performed.

Treatment: After the onset of the first clinical sign, the farm’s attending physician initiated intensive antimicrobial therapy with Shotapen^®^ (Virbac, Carros, France, a long-acting injectable penicillin–streptomycin solution in a standard therapeutic dose by IM injection) in combination with T.M.P.S.^®^ inj. (Kepro, Woerden, Netherlands, trimethoprim–sulfamethoxazole solution administered at a dosage of 15 mg/kg every 12 h by intravenous route). Due to the lack of improvement in the general condition, on the 3rd day, the treatment was continued with off-label use of Gentamox^®^ (Hipra, Amer (Girona), Spain. Amoxicillin–gentamicin, injectable solution in a standard therapeutic dose by IM injection). Supportive and symptomatic treatment included administering fluids, vitamins, and non-steroidal anti-inflammatory drugs (Phenylarthrite^®^, Vetoquinol, Magny-Vernois, France at dose of 2.2 mg/kg, SID for 3 days).

Outcome: Despite treatment, the clinical symptoms deteriorated, and no improvement in the patient’s general condition was noted over time. The filly died on 24 March 2025, at the age of 61 days. The illness lasted for 5 days, during which the foal exhibited signs of severe respiratory failure, such as shallow or irregular breathing and prolonged pauses between breaths. There were also observations of bluish or grayish discoloration of the mucous membranes and a loss of consciousness.

Gross pathology: A post-mortem examination was conducted, revealing suppurative bronchopneumonia with bilateral lung congestion and consolidation. There were multiple disseminated focal-to-coalescing nodules in the lung parenchyma, ranging in size from a few millimeters to 6 cm (Figure 1A,D). The nodules contained a white-yellowish substance that was visible on their cut surfaces (Figure 1B). A single nodular area of caseation, measuring 6.2 cm, was found in the cranioventral region of the right lung (Figure 1C). The findings also included pleural effusions, hemorrhages in the distal part of the trachea and bronchi andenlarged mediastinal lymph nodes.

Sampling: The lung with the adjacent lymph nodes was submitted to the microbiology laboratory of the Faculty of Veterinary Medicine at Trakia University (Stara Zagora, Bulgaria).

Other cases: The owner reported a similar case of illness in a foal of a comparable age that resulted in a fatal outcome three years ago. During the partial autopsy performed at the time, the attending veterinarian identified intussusception as the cause of death. However, a full autopsy was not conducted, and no samples were submitted for etiological diagnosis. According to farm records, respiratory disease has been detected in 3 out of 16 foals (aged up to 6 months) over the past 2 years. In all three cases, treatment was initiated promptly, and each case resulted in recovery.

Diagnosis: The preliminary diagnosis, based on the available evidence, including epidemiological data, clinical symptoms, and necropsy findings, included a bacterial infection with commonly reported pyogenic organisms such as *Streptococcus zooepidemicus*, *R. equi*, *Corynebacterium pseudotuberculosis*, *Actinobacillus equuli*, or others that are commonly reported as potential causes of similar conditions [9,10]. The epidemiological background, clinical presentation, and gross pathological findings were strongly indicative of *R. equi* infection, with no evidence suggesting alternative diseases. A conventional bacteriological examination, performed using enriched liquid and solid media, resulted in the isolation of an *R. equi* strain, while no other bacterial pathogens were detected. Based on the selective isolation and subsequent molecular confirmation of a VapA-positive *R. equi* strain, and in the absence of clinical or pathological features indicating other etiologies, alternative causative agents were considered unlikely.

Microbiological testing: Conventional microbiological tests were performed, including inoculations of liquid broth media (Soyabean Casein Digest Medium, HiMedia^®^, Thane, Maharashtra, India) with subsequent subcultures, as well as solid nutrient media (Blood Agar Base, HiMedia^®^, India, supplemented with 5% defibrinated bovine blood and MacConkey Agar w/0.15% Bile Salts, CV and NaCl, HiMedia^®^, India), incubated at 37 °C and under aerobic conditions for 7 days [11]. The initial confirmation of the isolates as *R. equi* was obtained through an analysis of culture characteristics, visualization of the typical cellular morphology in Gram-stained smears, and primary characterization tests. Bacteriological testing revealed that the isolate was catalase-positive, oxidase-negative, and incapable of fermenting carbohydrates. After 24 h of aerobic incubation on bovine blood agar at 37 °C, the translucent, nonhemolytic, smooth, and shiny colonies, which tended to merge due to strong mucus production, were barely noticeable (Figure 2A). After 48 h of cultivation, the isolate formed larger, smooth, irregularly round, mucoid colonies that appeared whitish on a solid medium. A close-up of the colonies showed hyphae-like structures (substrate mycelium or filamentous structures) surrounding the colonies (Figure 2B). After 72 h, the colonies displayed a pale pink color and a slimy texture (Figure 2C). The isolate was non-hemolytic but induced enhanced “spade-shaped” β-hemolysis of Staphylococcus aureus when cross-streaked—positive CAMP test (Figure 2D).

The cell morphology of *R. equi* during different growth phases shows that in the exponential phase, the cells are typically rod-shaped, whereas in the stationary phase, they tend to acquire a coccoid (spherical) shape (Figure 3).

Sensitivity to antimicrobials: The susceptibility to various antimicrobial agents was determined in vitro by the Kirby–Bauer disk diffusion technique on Mueller–Hinton medium enriched with bovine blood (HiMedia^®^, Maharashtra, India). After 24 h of incubation, the diameters of the zones of growth inhibition were measured. The results were interpreted using the three-point Kirby–Bauer scale and classified as susceptible (S), intermediate (I), or resistant (R) based on the current EUCAST and CLSI standards. Our findings showed that the isolate was sensitive to azithromycin, erythromycin, tulathromycin, clarithromycin, marbofloxacin, amoxicillin/clavulanic acid, streptomycin, and gentamicin. The strain demonstrated intermediate sensitivity to rifampicin, enrofloxacin, chloramphenicol, and trimethoprim–sulfonamides. However, it was resistant to penicillin, cephalexin, oxytetracycline, and clindamycin.

Identification of organism and detection of *VapA* virulence gene: A swab sample of cultivated bacteria was submitted to the commercial Laboratory Laboklin GmbH & Co. KG (Neckarsulm, Germany) for *R. equi* and *R. equi VapA* PCR. The swab sample was incubated for 20 min in cobas PCR media (Roche Diagnostics GmbH, Mannheim, Germany), and 200 μL of the supernatant was automatically pipetted by a Hamilton Microlab STARlet (Hamilton Germany GmbH—Robotics, Gräfelfing, Germany) into a MagNa Pure 96 Processing Cartridge (Roche Diagnostics GmbH, Mannheim, Germany). Automated total NA extraction was performed by the “MagNA Pure 96 DNA and Viral NA small volume kit” (Roche Diagnostics GmbH, Mannheim, Germany) according to the manufacturer’s instructions. The resulting NA was eluted in a volume of 100 μL. Isolated NA was stored at −20 °C. To control the successful NA extraction, an extraction control using the DNA or RNA Process Control (Detection) Kit (Roche Diagnostics GmbH, Mannheim, Germany) was included in every sample.

To prepare the master mixes for *R. equi* and *R. equi VapA*, the DNA Process Control Detection Kit (Roche Diagnostics GmbH) was used. Preparation of master mixes, mixture with NA, and the PCR itself were performed in separate rooms. Positive and negative controls were included in every PCR run.

Real-time PCR assays with hydrolysis probes (qPCR) were performed on a LightCycler^®^ 96 (Roche Diagnostics GmbH, Mannheim, Germany). The PCR program started with an initial denaturation step for 30 s at 95 °C, followed by 40 cycles of 5 s at 95 °C and 30 s at 60 °C. qPCRs were applied as qualitative assays (negative/positive). Samples were regarded as positive with a Cq-value of <35, evaluated at a threshold of 0.05 (Figure 4).

The following primers and probes for *R. equi (=R. hoagii)* and *R. equi (=R. hoagii) VapA* [12] were used: Rhod-F 5′-CGA CAA GCG CTC GAT GTG-3′, Rhod-R 5′-TGC CGA AGC CCA TGA AGT-3′and Rhod-probe 5′-FAM-TGG CCG ACA AGA CCG ATC AGC C-BHQ-3′ and vapA-R 5′-CAG CAG TGC GAT TCT CAA TAG TG-3′, vapA-F 5′-GAA GTC GTC GAG CTG TCA TAG CT-3′, and vapA-probe 5′-FAM-CAG AAC CGA CAA TGC CAC TGC CTG-BHQ-3′. **The** amplicon sizes were 68 bp for *R. equi* and 75 bp for *R. equi VapA*.

PCR tests confirmed the presence of *VapA*-positive *R. equi*.

## 3. Discussion

The case history data presents a sporadic, fatal respiratory infection in a young foal that represents the first proven case of a virulent strain of *R. equi* in Bulgaria.

The Rhodococcus bacterium is a ubiquitous soil microorganism that is predominantly found in the feces of herbivorous animals, especially horses. Takai [4] notes that despite its widespread environmental distribution, infections caused by the bacterium are endemic on certain farms, occur sporadically on others, and may remain undetected in the majority of equestrian facilities. Strains of *R. equi* exhibit variability in virulence; only those expressing the highly immunogenic surface protein called virulence-associated protein (*VapA*) are pathogenic to foals. In farms where the disease has not been diagnosed, low-level environmental contamination has been observed. In contrast, high frequencies of virulent strains have been reported in endemic farms, both in soil from pastures, within stables, and in the feces of foals [4,9,13].

The disease typically manifests as chronic bronchopneumonia with suppuration and abscess formation [14]. A less common form is the acute variant, lasting from a few hours to several days [15]. Rhodococcosis is either a sporadic or endemic disease, with the mortality rate ranging from 30% in foals suffering from pneumonia [16] to 57% in those who are affected by pneumonia and extrapulmonary disorders (EPDs) [17,18].

In this case the disease developed slowly over time, appeared suddenly, and progressed acutely over five days, with a gradual worsening of the filly’s condition despite treatment. It was manifested in a typical respiratory form, with severe pulmonary lesions and respiratory distress. EPDs were not observed, in contrast to many other outbreaks [18].

Although the disease has been known for a long time, many aspects of the infection remain unexplored. The distribution of virulent strains varies significantly between countries and even between farms within the same region. The emergence of *VapA*-positive strains, as observed in our case, aligns with reports from endemic areas such as the United Kingdom, Germany, France, and Spain, where both clinical and subclinical infections are increasingly detected through routine screening and ultrasonographic examination [4,13,15].

This case illustrates how individual detections can serve as warning events, drawing attention to potentially unrecognized endemic outbreaks or changing environmental conditions that favor transmission. Factors such as climate variability, intensive breeding practices, and increased horse trade between European Union Member States may facilitate the introduction and establishment of virulent strains in previously unaffected areas [14,19]. Furthermore, the detection of antimicrobial resistance trends across various European countries highlights the need for coordinated monitoring efforts. For example, isolates that are resistant to macrolides and rifampin are increasingly documented on endemic farms in France and Germany, posing a treatment challenge and necessitating stricter antimicrobial stewardship [5,20]. In our case, the isolate showed intermediate sensitivity to rifampin, which is a concerning finding in light of these broader resistance patterns. Incorporating this epidemiological perspective into national veterinary strategies could support the implementation of early diagnostic protocols, risk-based surveillance, and biosecurity measures tailored to local needs. The prevalence of the infection has been better understood with the introduction of ultrasonography as a screening tool. In horse breeding farms that utilize thoracic ultrasonography, it has been discovered that over 50% of foals exhibit lung lesions, such as parenchymal consolidation and abscess formation, without demonstrating any clinical symptoms. Among those infected, 20 to 25% develop clinical signs and require antibiotic treatment [13]. This means that multiple factors influence the levels of clinical manifestation and vary across horse farms. Disease rates can be affected by several factors, including the levels of environmental contamination, the presence of a high number of susceptible foals, colostrum management, and other conditions that lead to a decrease in the overall resistance of the foals. One of the key factors in improving neonatal protection is ensuring timely intake of colostrum, as the epitheliochorial placentation in mares prevents the transplacental transfer of protective maternal antibodies to the fetus [21]. While some studies confirm the importance of colostrum intake in protecting foals against *R. equi* [22], others have found it insufficient for complete protection [23]. In our case, despite the timely intake of an adequate amount of colostrum, it was insufficient to prevent the disease and its fatal outcome. A possible explanation is provided by a study by Gardner et al. [24], who found that hospitalized horses exhibited lower phagocytic activity and opsonizing capacity than normal. The study suggests that there is not necessarily a correlation between IgG levels and opsonizing capacity in some foals.

Key factors for successful disease control include timely diagnosis, appropriate treatment of affected animals, and maintaining environmental hygiene to reduce the risk of infection [15]. The most recommended antibiotics for treatment are macrolides (erythromycin, clarithromycin, or azithromycin) in combination with rifampin. These antibiotics are lipophilic, allowing them to penetrate affected tissues and macrophages, where they exhibit bacteriostatic or bactericidal action depending on their concentration [5]. Macrolide antibiotics possess a positive charge in aqueous solutions, especially at physiological pH levels. This positive charge is essential for their interactions with bacterial ribosomes, as well as with lipids and other biological macromolecules. Macrolides are classified as weak bases and exhibit lipophilic characteristics, with a relatively high molecular weight of approximately 500 Daltons. They tend to be unstable in acidic environments [25]. However, newer derivatives like azithromycin and roxithromycin demonstrate improved stability in acidic conditions [26]. Azithromycin has an advantage because pus is acidic, with a pH range of 5.5 to 7.2 [27]. However, these antibiotics are expensive and not readily available in practice, leading veterinarians to often initiate treatment with other more accessible agents, which may be less effective, as in our case. An increasing problem is the development of resistance of *R. equi* to macrolides and rifampin on some farms, complicating treatment [5]. In our case, the isolate showed intermediate sensitivity to rifampin, which is particularly concerning. Therefore, it is important to conduct sensitivity tests on all isolates to monitor resistance and select appropriate treatment. Due to the intracellular localization and sequestration nature of the inflammation, there is often a discrepancy between the in vitro activity of antimicrobial agents and their in vivo efficacy, posing another challenge in selecting antimicrobial agents [5].

Berghaus et al. [28] found that clarithromycin was more active than azithromycin, erythromycin, and gamithromycin against intracellular *R. equi*, while Giguère et al. [29] reported that enrofloxacin, gentamicin, and vancomycin were significantly more active than other drugs, with doxycycline being the least active. Some studies have demonstrated the effective use of weekly tulathromycin at a dose of 2.5 mg/kg body weight, administered intramuscularly [30,31]. A similar study in Germany showed results that were comparable to those of azithromycin and the combination of azithromycin with rifampin [20]. The duration of therapy, which generally ranges between 3 and 12 weeks, depends on the severity of the initial lesions and the clinical response [5]. Despite the availability of active antibiotics and long-term therapeutic courses, the prognosis is not very encouraging. Giguére [5] summarizes that before the introduction of the combination of erythromycin and rifampin as the treatment of choice in the early 1980s, the prognosis of infected foals was poor, with survival rates as low as 20%. Later, it rose to 88% using erythromycin and rifampin [32]. It must be emphasized that while in the treatment of clinically manifest disease, the effectiveness varies between 59% and 72% in some studies, in farms implementing a screening program to identify and treat foals with subclinical lesions, the survival rate approaches 100% [5].

The filly was treated by the attending veterinarian with a combination of penicillin–streptomycin, followed by amoxicillin–gentamicin, but there was no improvement. This approach is commonly used in practice to treat respiratory infections; however, it is often flawed or suboptimal. While in vitro susceptibility testing might indicate some effectiveness, the in vivo conditions frequently render these antibiotics clinically ineffective. In this regard, penicillins and aminoglycosides possess low molecular weights, which facilitate their diffusion toward areas of lower concentration. However, their classification as hydrophilic compounds impedes this diffusion process [27]. The thick, encapsulated abscesses that are typical of *R. equi* create an acidic environment that further impairs the efficacy of these agents [27]. Many authors have claimed in the past that aminoglycosides either do not penetrate mammalian cells or penetrate poorly. However, in vivo studies have indicated that the primary mechanism for the uptake of aminoglycosides is endocytosis. Notably, these compounds are not distributed uniformly within the cells; instead, they predominantly localize within lysosomes [33]. The effectiveness of penicillins against intracellular bacteria is regarded as limited. Studies have shown that penicillin antibiotics also have poor cell penetration [34,35]. This discrepancy highlights the importance of choosing antimicrobials with proven intracellular penetration, such as macrolides combined with rifampin, which remain the standard of treatment.

To prevent therapeutic failures in future cases, veterinarians should support their diagnoses with thoracic ultrasonography, blood tests, and microbiological studies, along with susceptibility testing. If there is reasonable suspicion, they should initiate appropriate treatment. Additionally, creating farm-specific treatment protocols based on local resistance patterns and improving access to effective antibiotics could enhance therapeutic outcomes. The prolonged, expensive, and uncertain treatment raises the question of disease prevention. Unfortunately, there is still no effective commercial vaccine available, despite recent progress in vector-based vaccines [19]. On affected farms, it is recommended to implement early detection and monitoring, targeted treatment, biosecurity, and environmental management. It is also important to manage certain factors effectively, such as preventing overcrowding, ensuring housing in well-ventilated and dust-free areas, minimizing dirt or dusty surfaces, rotating pastures, and isolating foals that show clinical signs of illness [36]. Foals should be closely monitored for early recognition of disease. This monitoring should include daily rectal temperature checks, serologic surveillance, measuring plasma fibrinogen levels every two weeks, and conducting physical examinations, ultrasonography, or X-rays as needed [37].

An important part of control is also the disinfection of the environment with appropriate disinfectants [38], especially given the high levels of shedding in feces [15,39], exhaled air, and respiratory secretions [40]. Contaminated soil plays a significant role—*R. equi* is often defined as a coprophilic soil-associated actinomycete [41]. Comprehensive farm management practices could potentially reduce the risk of pathogen spread and the occurrence of infections in susceptible foals, other animal species, and humans.

## 4. Conclusions

This report describes the first confirmed case of *R. equi* infection in horses in Bulgaria and contributes to our understanding of its widespread distribution. This finding should alert both owners and veterinarians to potential health issues in foals on breeding farms. Rhodococcus infection has a prolonged subclinical phase, and its clinical symptoms can be indistinguishable from those caused by other pneumonic bacteria. Additionally, this infection requires specific medications and an extended course of treatment. The epidemiological significance of this particular case is crucial, as foals with pneumonia are the primary source of virulent *R. equi* contamination in the environment. Timely diagnostic measures, such as ultrasonography, bloodwork, and a thorough clinical examination, are essential to accurately assess each case of disease. To prevent permanent environmental contamination by virulent strains, it is essential to isolate sick foals, commence disease-specific therapy, and routinely sanitize and disinfect the surroundings. Ultimately, this will impact the incidence of infections and, as a result, decrease the overall use of antibiotics.

## Figures and Tables

**Figure 1 animals-15-02058-f001:**
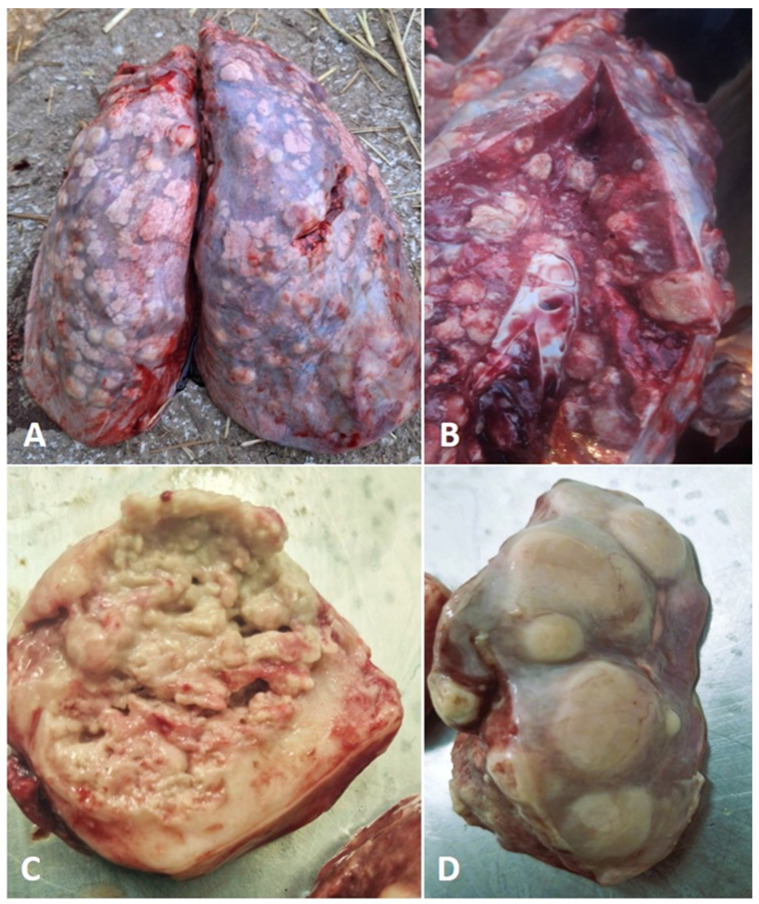
Gross pathological picture of lesions. (**A**) Disseminated nodules in the lung parenchyma; (**B**) white-yellowish substance on the nodular cut surface; (**C**) caseation of the nodular area; (**D**) nodules with thick-walled capsules.

**Figure 2 animals-15-02058-f002:**
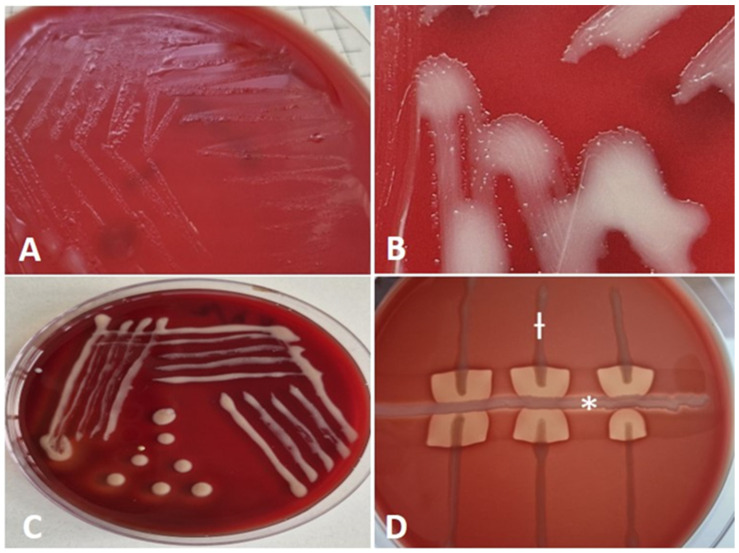
Culture of *R. equi* inoculated on bovine blood agar 5% at 37 °C. (**A**) Translucent, smooth, shiny, nonhemolytic colonies after 24 h incubation; (**B**) hyphae-like structures after 48 h incubation; (**C**) mucoid, teardrop, coalescing, glistening salmon-colored colonies after 72 h incubation; (**D**) positive CAMP test of *R. equi* (ƚ) with *S. aureus* (*).

**Figure 3 animals-15-02058-f003:**
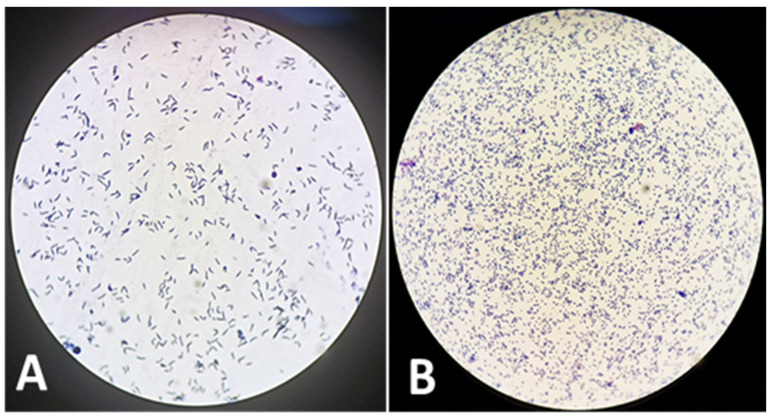
Microscopic images illustrating the differential cell morphology of the isolate observed in the exponential (**A**) and stationary (**B**) growth phases.

**Figure 4 animals-15-02058-f004:**
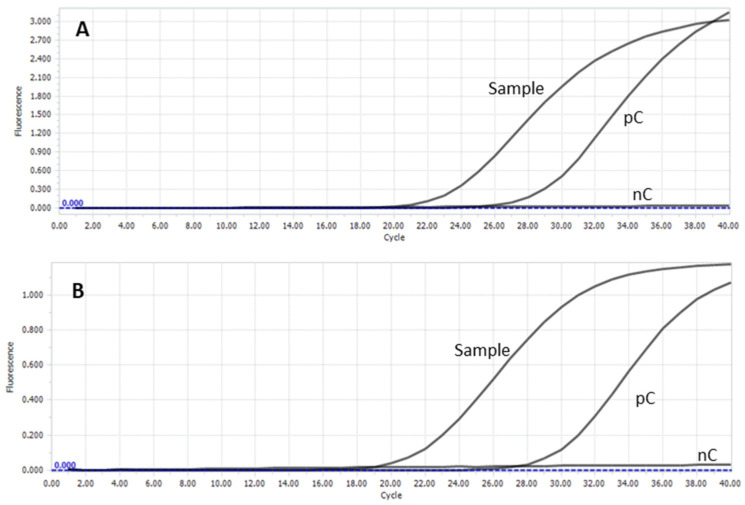
Results of real-time PCR for *R. equi* (**A**) and *R. equi VapA* (**B**). pC: positive control; nC: negative control. *y* axis: Fluorescence; *x* axis: Cycle. Cq value for *R. equi*: 20.83; Cq value for *R. equi VapA*: 20.45.

## Data Availability

The original contributions presented in this case are included in the article. Further inquiries can be directed to the corresponding author.

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
