# Peer review of "First Detection of Rhodococcus equi in a Foal in Bulgaria—A Case Report"

_animals, 2025, doi:10.3390/ani15142058_

Round 1
Reviewer 1 Report
Comments and Suggestions for Authors
The manuscript is clinically relevant and provides valuable new insights for veterinary practitioners. However, a significant limitation is that it is based solely on a single case report. To enhance the manuscript, the authors should substantially broaden the discussion, specifically addressing the implications of their findings on the epidemiology of Rhodococcus equi infections across Europe. Additionally, it would be beneficial to critically discuss the rationale behind the selected antimicrobial therapy, reasons for its ineffectiveness, and strategies to avoid treatment failures in similar cases.
Minor comments:
The correct species name should be Rhodococcus equi, as supported by recent literature (https://pubmed.ncbi.nlm.nih.gov/37796005/; https://pubmed.ncbi.nlm.nih.gov/32375930/). The gene name vapA should consistently be italicized (e.g., lines 2, 18, etc.). Similarly, all species names must be italicized throughout the manuscript (e.g., line 35). The wording on line 38 should be revised to adopt a more formal scientific style. Latin expressions such as in vitro and in vivo must also be italicized (e.g., lines 38 and 39). Finally, ensure that the first mention of a bacterial species includes the full genus and species names, while subsequent mentions can use the abbreviated form (e.g., R. equi).
Reviewer 2 Report
Comments and Suggestions for Authors
The authors have documented the first report of VapA-positive Rhodococcus hoagii (formerly known as Rhodococcus equi) infection in a foal in Bulgaria. The following comments are provided for consideration:
- The title of the manuscript may be revised. Please consider whether it is necessary to include "VapA-positive" in the title.
- Please include the most recent classification of Rhodococcus hoagii infection in foals.
- Please justify the importance of the VapA gene in Rhodococcus infections in foals.
- Briefly include the prevalence of respiratory tract infections on the farm within the manuscript.
- If available, the manuscript should be supported with histopathological figures.
- The authors should clearly describe how other etiological agents were ruled out. This should be supported by both conventional and molecular test results.
- Please include the PCR image confirming VapA-positive Rhodococcus hoagii infection in the foal.
- Provide details of the primers and probes used for both conventional and real-time PCR analysis, indicating whether they were previously published or newly designed by the authors.
- PCR conditions and amplicon sizes are currently missing and should be included in the manuscript.
- Real-time PCR analysis should be supported with relevant data and figures.
- The discussion section should be revised to clearly describe the treatment protocol followed, if any.
- The manuscript should be thoroughly checked for grammatical and typographical errors.
- References should be formatted in accordance with the journal’s author guidelines.
Round 2
Reviewer 1 Report
Comments and Suggestions for Authors
No further comments.